# Temperature Control of Spring $CO_2$ Fluxes at a Coniferous Forest and a Peat Bog in Central Siberia

**Sung-Bin Park** [1,*], **Alexander Knohl** [2,3], **Mirco Migliavacca** [1], **Tea Thum** [1], **Timo Vesala** [4,5,6], **Olli Peltola** [7], **Ivan Mammarella** [4], **Anatoly Prokushkin** [8], **Olaf Kolle** [1], **Jošt Lavrič** [1], **Sang Seo Park** [9] and **Martin Heimann** [1,4]

1   Max Planck Institute for Biogeochemistry, Hans-Knöll-Street10, 07745 Jena, Germany; mmiglia@bgc-jena.mpg.de (M.M.); tea.thum@fmi.fi (T.T.); okolle@bgc-jena.mpg.de (O.K.); jlavric@bgc-jena.mpg.de (J.L.); martin.heimann@bgc-jena.mpg.de (M.H.)
2   Bioclimatology, Faculty of Forest Science and Forest Ecology, University of Göttingen, Büsgenweg 2, 37077 Göttingen, Germany; aknohl@uni-goettingen.de
3   Centre of Biodiversity and Sustainable Land Use (CBL), University of Göttingen, Büsgenweg 1, 37077 Göttingen, Germany
4   Institute for Atmospheric and Earth System Research (INAR)/Physics, Faculty of Sciences, University of Helsinki, P.O. Box 64, 00014 Helsinki, Finland; timo.vesala@helsinki.fi (T.V.); ivan.mammarella@helsinki.fi (I.M.)
5   Institute for Atmospheric and Earth System Research (INAR)/Forest Sciences, Faculty of Agriculture and Forestry, University of Helsinki, Viikinkaari 1, 00014 Helsinki, Finland
6   Yugra State University, 628012 Khanty-Mansiysk, Russia
7   Climate Research Programme, Finnish Meteorological Institute, P.O. Box 503, 00101 Helsinki, Finland; olli.peltola@fmi.fi
8   Vladimir Nikolayevich Sukachev Institute of Forest of the Siberian Branch of Russian Academy of Sciences, Separated Department of the KSC SB RAS, 660036 Krasnoyarsk, Russia; prokushkin@ksc.krasn.ru
9   School of Urban and Environmental Engineering, Ulsan National Institute of Science and Technology (UNIST), 50, UNIST-gil, Ulsan 44919, Korea; sangseopark@unist.ac.kr
*   Correspondence: sungbina@unist.ac.kr or sungbin.beaniya@gmail.com

**Abstract:** Climate change impacts the characteristics of the vegetation carbon-uptake process in the northern Eurasian terrestrial ecosystem. However, the currently available direct $CO_2$ flux measurement datasets, particularly for central Siberia, are insufficient for understanding the current condition in the northern Eurasian carbon cycle. Here, we report daily and seasonal interannual variations in $CO_2$ fluxes and associated abiotic factors measured using eddy covariance in a coniferous forest and a bog near Zotino, Krasnoyarsk Krai, Russia, for April to early June, 2013–2017. Despite the snow not being completely melted, both ecosystems became weak net $CO_2$ sinks if the air temperature was warm enough for photosynthesis. The forest became a net $CO_2$ sink 7–16 days earlier than the bog. After the surface soil temperature exceeded ~1 °C, the ecosystems became persistent net $CO_2$ sinks. Net ecosystem productivity was highest in 2015 for both ecosystems because of the anomalously high air temperature in May compared with other years. Our findings demonstrate that long-term monitoring of flux measurements at the site level, particularly during winter and its transition to spring, is essential for understanding the responses of the northern Eurasian ecosystem to spring warming.

**Keywords:** spring; eddy covariance; $CO_2$ flux; temperature; snowmelt; boreal forest; peatland; Siberia; carbon cycle; northern Eurasia

## 1. Introduction

Boreal forests and peatlands are the major terrestrial biomes and large carbon (C) reservoirs in northern Eurasia, occupying 49% and 25% of Russia's land area, respectively [1,2]. Both ecosystems are considered essential C sinks in the global C cycle [2–8]. The role of annual or seasonal C-sink capacity in boreal forests can vary depending on temperature

and environmental factors (e.g., age, management) [5,9,10]. In case of peatlands, quantifying the seasonal C-uptake capacity seems more complex than in forests because of various ecohydrological factors (e.g., vegetation, soil microbes) [11–14]. Considering only $CO_2$ exchange processes, both boreal forests and peatlands in northern Eurasia are generally considered as net $CO_2$ sinks during the snow-free season [15–18].

During the seasonal transition from winter to spring, rapidly increasing radiation, temperature, and water availability after snowmelt affect plant photosynthesis and associated processes, thereby impacting the net ecosystem exchange of $CO_2$ (NEE). For instance, gross $CO_2$ uptake rates gradually increase from April to May owing to the increase in photochemical efficiency and more favorable meteorological conditions during that period [19,20]. The photosynthetic apparatus of boreal forests is adapted to quickly respond to positive temperatures and spring snowmelt, resulting in rapid recovery of physiological activity [20–24]. Similarly, a reactivation of photosynthesis for peat mosses occurs immediately after they are exposed from the snow cover [18,25–27]. At the beginning of snowmelt, the surface soil temperature exceeds 0 °C but remains close to 0 °C until snowmelt completion [21,25]. Once the snow has melted, surface soil temperature increases rapidly, and its diurnal cycle becomes pronounced [28]. Overall, the above mentioned studies demonstrate that the photosynthetic capacity of both boreal forests and peatlands are strongly influenced by the interannual variability in abiotic and environmental conditions during spring. Therefore, $CO_2$ flux data for the winter-to-spring transition period are crucial to understanding the C cycle in northern Eurasia.

Over the past five decades, substantial air temperature warming during spring has reduced the extent of snow cover in northern Eurasia [4,29–33]. A study by Pulliainen et al. [22] showed that during 1979–2014, earlier snowmelt induced by temperature warming increased spring ecosystem productivity in boreal forests. Earlier snowmelt impacts the earlier start of phenological development and may result in enhanced vegetation productivity [30]. However, other studies have shown that since the 2000s, the temperature-warming trend has slowed down in high-northern latitudes [31]. A slowed temperature increase may weaken C uptake in boreal forests [32]. Moreover, snow-cover reduction in northern Eurasia differs depending on the considered period or region [33]. These variations imply a need for continuous monitoring of the $CO_2$ flux at the ecosystem scale in northern Eurasia, particularly during winter and the transition to spring.

Eddy covariance (EC) flux measurements can provide direct information about biosphere-atmosphere interaction and photosynthesis-related processes at the ecosystem scale [34,35]. Boreal coniferous forests and peat bogs are major biomes in the Russian middle taiga [36]. However, EC flux data are still sparse for Siberia [37]. This lack of data motivated the first initiative of EC measurements in Siberia during 1999–2003 [15,38]. Various studies have quantified seasonal and inter-annual variabilities in $CO_2$ fluxes in Siberian boreal forests [15,28,36]. In addition, comparisons of $CO_2$ fluxes over the different peatland types have been reported [38–40]. In particular, Arneth et al. [21] highlighted differences in carbon- and energy-flux dynamics between the Scots pine forest and peat bog in central Siberia for April to May 1999–2000. However, no EC flux measurements were performed for the period of 2003 to mid-2012 in this area. To quantify the long-term biogeochemical cycle in northern Eurasia, EC flux measurements were subsequently re-initiated in a coniferous forest and a bog adjacent to the Zotino Tall Tower Observatory (ZOTTO) [40]. $CO_2$ flux measurements collected after mid-2012 were reported by Winderlich et al. [41] and Park et al. [42]; however, these studies analyzed the 2012–2013 growing season only. Despite the importance of the snow to the snow-free period in influencing the C cycle, to our best knowledge, there has been no subsequent investigation highlighting springtime after the study of Arneth et al. [21] in central Siberia.

Here, we report five years of springtime $CO_2$ flux data (April to early June 2013–2017) near Zotino in Russia. Our objectives are: (a) to characterize the difference in seasonal $CO_2$ fluxes and their responses to abiotic factors between the two ecosystems and among years; (b) to identify the factors explaining the $CO_2$ fluxes in a coniferous forest and a bog

in spring; and (c) to examine the effects of temperature on the strength of net $CO_2$ sink and the timing of the start of $CO_2$ uptake. In particular, we focus on the seasonal variations of $CO_2$ fluxes for the three years (2014–2016). Lastly, we discuss the effect of spring warming on snowmelt and the ability of net $CO_2$ sink in both ecosystems.

## 2. Materials and Methods

### 2.1. Study Site

The Zotino forest flux tower (hereafter ZF; 60°48′25″ N, 89°21′27″ E, elevation 110 m a.s.l.) is situated 900 m to the north-northeast of the ZOTTO (Figure 1). The average canopy height of the forest is approximately 20 m, and the measurement height is 30.3 m (Table 1). The dominant tree species is Scots pine (*Pinus sylvestris*), ranging in age from 80 to 180 years, whereas the patchy distributed regrowth Scots pine (height < 5 m) in understory is represented by younger pine trees of several age groups (40 years). The main ground vegetation within the footprint area is lichen (*Cladina stellaris* and *Cladina rangiferina*), with patches of dwarf shrub (*Vaccinium vitis-idaea*).

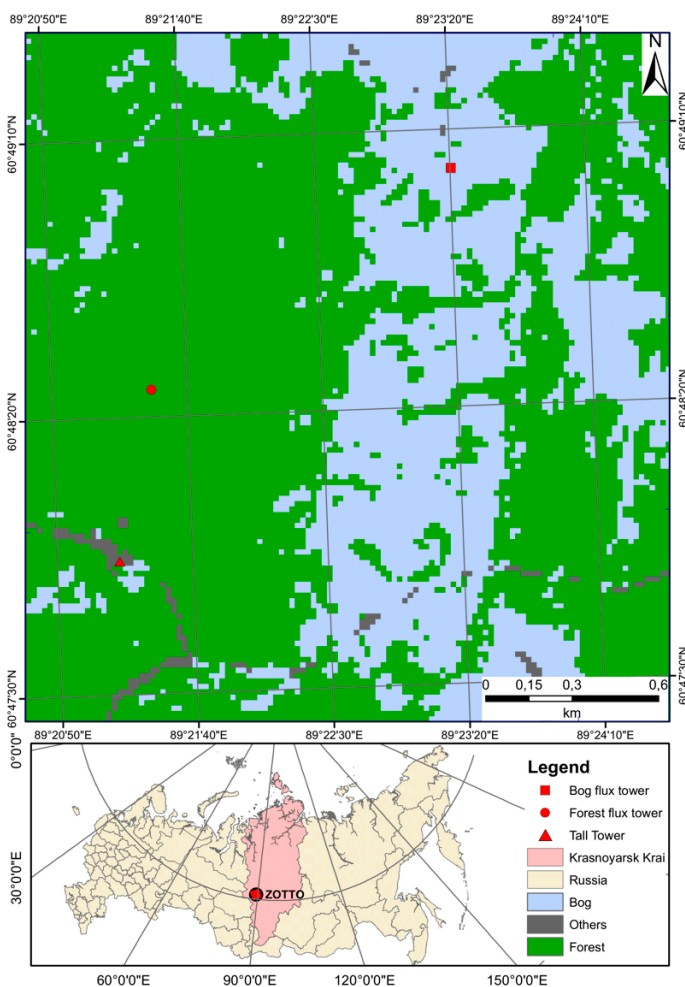

**Figure 1.** Land-cover map and geographical locations of forest (ZF; red circle) and bog (ZB; red rectangle) flux towers and tall tower (red triangle) in Zotino, Krasnoyarsk Krai, Russia (black circle). The map is derived from 30 m resolution Landsat-8 image. The ZOTTO denotes the Zotino Tall Tower Observatory. We reclassified 14 initial land-cover types into 3 types. Forest includes reforestation, regrowth, and lichen with pines. Bog includes shrubs, flooded, and wet body. Other classes (others) include clear-cut/barren, burned, sand, and sparse vegetation.

**Table 1.** Site characteristics of the forest and bog flux towers at Zotino.

|  | ZF | ZB |
|---|---|---|
| Geographical location | 60.48′25″ N 89.21′27″ E | 60.49′03″ N 89.23′20″ E |
| Elevation altitude (m a.s.l) | 110 | 66 |
| Vegetation type | Coniferous forest | Ombrotrophic bog |
| Plant species | *Pinus sylvestris* | *Sphagnum* |
| Vegetation height (m) | 20 for dominant trees 5 for understory | 2.5 for Scots pines 0.5 for dwarf shrubs |
| Measurement height (m) [1] | 29.7 | 10.1 |
| Tower height (m) [2] | 29.4 | 9.8 |
| Zero plan displacement [3] height (m) | 13.4 | 1.675 for pine trees 0.335 for dwarf shrubs |
| Roughness length (m) [4] | 3 | 0.375 for pine trees 0.075 for dwarf shrubs |
| Leaf area index (m$^{-2}$ m$^{-2}$) | 1–3.5 [5] | not available |
| Soil type | Podzol | Histosol |

[1] Measurement height was taken as the height measured to the centre of the sonic anemometers. [2] Tower heights were taken as the distance from the ground fundament panels to the top of the tower plate. [3] Zero plan displacement height was computed as 0.67 × vegetation height. [4] Roughness length was computed as 0.15 × vegetation height. [5] Values obtained from the inventory and remote-sensing measurements [43,44].

The Zotino bog flux tower (hereafter ZB, 60°49′03″ N, 89°23′20″ E, 66 m a.s.l.) is situated approximately 2 km to the northeast of the ZOTTO. The average canopy height at the site was approximately 2.5 m, and the measurement height was 9.9 m (Table 1 and Figure 1). The type of peat is classified as an ombrotrophic [21]. The landscape was covered by a pine-dwarf, shrub-sphagnum (in Siberia, called *ryam*), hollow-ridge complex. They had a height of 0.4–0.6 m and were covered by plant communities consisting of dwarf pine (*Pinus sylvestris f. litwinowii*), which dominates the trees and dwarf shrubs (*Chamaedaphne calyculata*). The peat's calibrated age at the bottom of the bog ranged from 9397 ± 134 y BP at the edges to 13617 ± 190 y BP at the bog's center. The peat depth showed a wide range from 1.60 m to 5.10 m, increasing toward the bog's center.

*2.2. Measurement System*

Identical micrometeorological measurement systems were installed at the forest and bog sites (Table 2). The EC system consisted of a three-dimensional ultrasonic anemometer (USA-1, Metek GmbH, Elmshorn, Germany) with integrated 55 W heating and closed-path infrared gas analyser (LI-7200, LiCor Biosciences, Lincoln, NE, USA) to measure $CO_2$ and $H_2O$ fluxes at 20 Hz frequency. An external diaphragm vacuum pump transported air to the gas analyser with a flow rate of 13 L min$^{-1}$ at ambient atmospheric pressure. At the top of the towers, sensors measured the four radiation components and photosynthetically active radiation (PAR), air temperature ($T_a$), relative humidity, and atmospheric pressure. Soil or peat temperature ($T_s$) was measured by PT100 soil-temperature probes at six depths (0.02, 0.04, 0.08, 0.16, 0.32, and 0.64 m for forest and 0.04, 0.08, 0.16, 0.32, 0.64, and 1.28 m for bog, respectively). Soil moisture was measured by six sensors at both sites: two sensors at 0.08 m and one sensor at each depth of 0.16, 0.32, 0.64, and 1.28 m at the ZF and six sensors at a depth of 0.08 m at the ZB. Data collected from the EC system and meteorological measurements were stored on a data logger (CR3000, Campbell Scientific Inc., Logan, UT, USA). Details in setup of the EC system have been described by Park et al. [42]. Snow depths at both sites were measured manually at locations nearby the towers; however, the measurement intervals were irregular.

**Table 2.** Instrument setup and sensor types of the flux towers at the Zotino sites.

|  | ZF | ZB |
|---|---|---|
| Sonic anemometer | USA-1, METEK GmbH, Elmshorn, Germany | |
| $CO_2/H_2O$ gas analyser | LI-7200, LiCor Biosciences, Lincoln NE, USA | |
| Time lag | 0.8 ($CO_2$), 1.2 ($H_2O$) | 0.9 ($CO_2$), 1.3 ($H_2O$) |
| Flow rate (L/min) | 13 | |
| Sampling frequency (Hz) | 20 | |
| Long/short wave up/downwelling radiation | CNR1, Kipp & Zonen, Deft, The Netherlands | |
| Up/downward photosynthetically active radiation | PQS1, Kipp & Zonen, Deft, The Netherlands | |
| Air temperature and relative humidity | KPK 1/6-ME-H38, Mela, Bondorf, Germany | |
| Barometric pressure | Pressure Transmitter, 61302 V, R.M. Young Company, Traverse City, USA | |
| Soil temperature | RTD temperature probe, Pt100, JUMO | |
| Soil moisture | ThetaProbe ML2x, Delta-T devices, Cambridge, England | |
| Ground heat flux | Heat flux plates RIMCO HP3/CN3, McVan Instruments, Mulgrave, Victoria, Australia | |
| Precipitation | Tipping bucket rain gauge, Adolf Thies GmbH & Co. KG, Göttingen Germany | |

### 2.3. Post-Processing of Data and Quality Control

We applied the data processing scheme for the closed-path analysers mentioned in Mammarella et al. [45]. In the raw data post-processing step, spike detection, 2D coordinate rotation, and time lag adjustment as well as calculation of half-hourly turbulent fluxes were performed using the EddyUH software [45]. The thresholds of friction velocity (u*) for ZF and ZB were 0.2 m s$^{-1}$ and 0.1 m s$^{-1}$, respectively. Details of the post-processing steps and quality control have been described by Park et al. [42]. The high-quality data available after the quality check and u* filtering led to a data coverage of 42–61% for ZF and 19–51% for ZB for 2013–2017. To calculate the cumulative $CO_2$ flux (NEE$_{cum}$), we gap-filled the half-hourly data using the REddyProc [46] in R [47]. Storage flux calculation was applied only for ZF data.

### 2.4. Data Anlaysis and Statistical Model

In this study, 'spring' is defined from the day of year (DOY) of 91–165 (01.04–14.06), 2013–2017. Daily $CO_2$ fluxes were summed for each 24-h period with gap-filled data. Regarding the EC convention, a negative NEE or $CO_2$ flux refers to net $CO_2$ uptake by the ecosystem, whereas a positive NEE indicates $CO_2$ release from the ecosystems to the atmosphere.

We used surface albedo (Alb) as a proxy for the status of snowmelt, as suggested in previous studies [22]. We determined the final day of snowmelt on which the 3-day moving average of Alb fell below 0.15 (15%) for the first time. To detect the final day of snowmelt, Shibistova et al. [28] used the daily mean soil temperature at a depth of 0.05 m and a 15-day moving average as the final day of snowmelt; however, in the present study, we used diurnal patterns of surface or peat temperature at a depth of 0.04 ($T_{s04}$). To examine the effect of temperature accumulation on vegetation productivity, we defined the accumulated growing degree days for air temperature (CGDD$_{Ta}$) and surface soil or peat temperature (CGDD$_{Ts04}$). We computed the cumulative sums of daily mean $T_a$ and $T_{s04}$ above 0 °C (base temperature 0 °C) from DOY 99–165.

To identify the abiotic variables that are important in explaining the variability in daily $CO_2$ flux, we used the multivariate adaptive regression splines (MARS) regression model [48], specifically, the earth package [49] in R [47]. Daily $CO_2$ fluxes with corresponding abiotic variables were used as training data for both sites from DOY of 99 to 165 in 2013–2017 for ZF and the same days in 2014–2016 for ZB, respectively. Several previous

studies have shown that spring $CO_2$ uptake is associated with four environmental factors: air temperature ($T_a$), $T_{s04}$, beginning day of snowmelt, and final day of snowmelt [21–24]. Therefore, we included $T_a$, $T_{s04}$, and Alb to construct the MARS model. For ZF, we used nine abiotic variables as training data: PAR, Alb, $T_a$, midday vapour pressure deficit (VPD), $T_{s04}$, and soil or peat temperature at depths of 0.08 m ($T_{s08}$), 0.16 m ($T_{s16}$), 0.32 m ($T_{s32}$), and 0.64 m ($T_{s64}$). For ZB, a total of seven abiotic variables were used as training data: the same as those for ZF but excluding $T_{s08}$ and $T_{s16}$ due to the long-term data gaps (>3 months) in the data for these two variables.

The MARS is a non-parametric regression method that can deal with both linear and nonlinear relationships and interactions between variables in the data using hinge functions [48]. Variable selection algorithms search the variables using both forward and backward stepwise selections. Variable importance is determined by a selection algorithm and is based on the number of model subsets (nsubsets), generalized-cross validation (GCV) score, and residual sum of squares (RSS). The GCV and RSS were scaled from 0 to 100. Higher GCV scores indicate variables with more explanatory power. The largest summed decrease of the RSS is scaled to 100, meaning that a large net decrease in the RSS is more important than a low RSS. Higher nsubsets means that variables are included in more subsets because they are important. Further statistical theory and application are described in detail at http://www.milbo.org/doc/earth-notes.pdf (accessed on 26 July 2021).

To investigate the impacts of cold spring weathers on the net ecosystem productivity (NEP), we used a rectangular hyperbolic light-response functional relationship [50] between PAR and NEP. We examined the differences in curve shapes and fit parameters for the year with the most frequent cold spring weather days (2014) and the year with the least frequent cold spring weather days (2015) with the following Equation (1):

$$\text{NEP} = \frac{\alpha * A_{max} * \text{PAR}}{A_{max} + \alpha * \text{PAR}} + R_d \tag{1}$$

where PAR ($\mu$mol photon m$^{-2}$ s$^{-1}$) is the incident photosynthetically active radiation, $A_{max}$ ($\mu$mol $CO_2$ m$^{-2}$ s$^{-1}$) is the light-saturation point of $CO_2$ uptake, and $\alpha$ is the initial slope of the NEP-PAR (net $CO_2$ uptake at light saturation), and $R_d$ is the ecosystem respiration during the day ($\mu$mol $CO_2$ m$^{-2}$ s$^{-1}$). We used daytime data for which potential global radiation ($R_{pot}$) exceeded 20 W m$^{-2}$. To make it easier to visualize, we used NEP, which is opposite in sign to NEE. Three model parameters were estimated for the selected two periods estimated using the Levenberg–Marquardt method, implemented in the minpack.lm package [51] in R [47]. The Levenberg–Marquardt method is a widely used algorithm for analyzing non-linear light-response curves [52,53]. $\text{NEP}_{sat}$ was calculated with the obtained three parameters and by fixing PAR at 1500 $\mu$mol photon m$^{-2}$ s$^{-1}$. The definition of $\text{NEP}_{sat}$ was the similar concept addressed by Migliavacca et al. [52] and Musavi et al. [54]. $\text{NEP}_{sat}$ helps to understand NEP at light saturation when PAR as 1500 $\mu$mol photon m$^{-2}$ s$^{-1}$ corresponds to light saturation.

## 3. Results

### 3.1. Abiotic Controls of Spring $CO_2$ Fluxes

Temporal evolutions of daily $CO_2$ fluxes and associated abiotic drivers in 2016 showed that $CO_2$ flux variations generally corresponded to the rapid increasing $T_a$ and $T_{s04}$ after snowmelt had completed (Figure 2). We confirmed that these highly fluctuating features of fluxes and spring meteorological conditions were similar in other years (Supplementary Figure S1). $CO_2$ fluxes at ZF increased faster than at ZB after snowmelt. Similar features were observed in other years. Alb at ZB was remarkedly different from that at ZF because of the higher radiation absorption by forest. Sporadic warm spells temporarily led to a rapid net uptake of $CO_2$ at ZF. In contrast, net $CO_2$ uptake rates at ZB did not increase as rapidly as those at ZF because the *Sphagnum* peat was still under snow cover and therefore less affected by changes in $T_a$. During the snowmelt period (DOY 127–145), the forest

transitioned from a net $CO_2$ source to a net $CO_2$ sink. However, $CO_2$ fluxes were still highly variable during this time, fluctuating between net $CO_2$ source and net $CO_2$ sink.

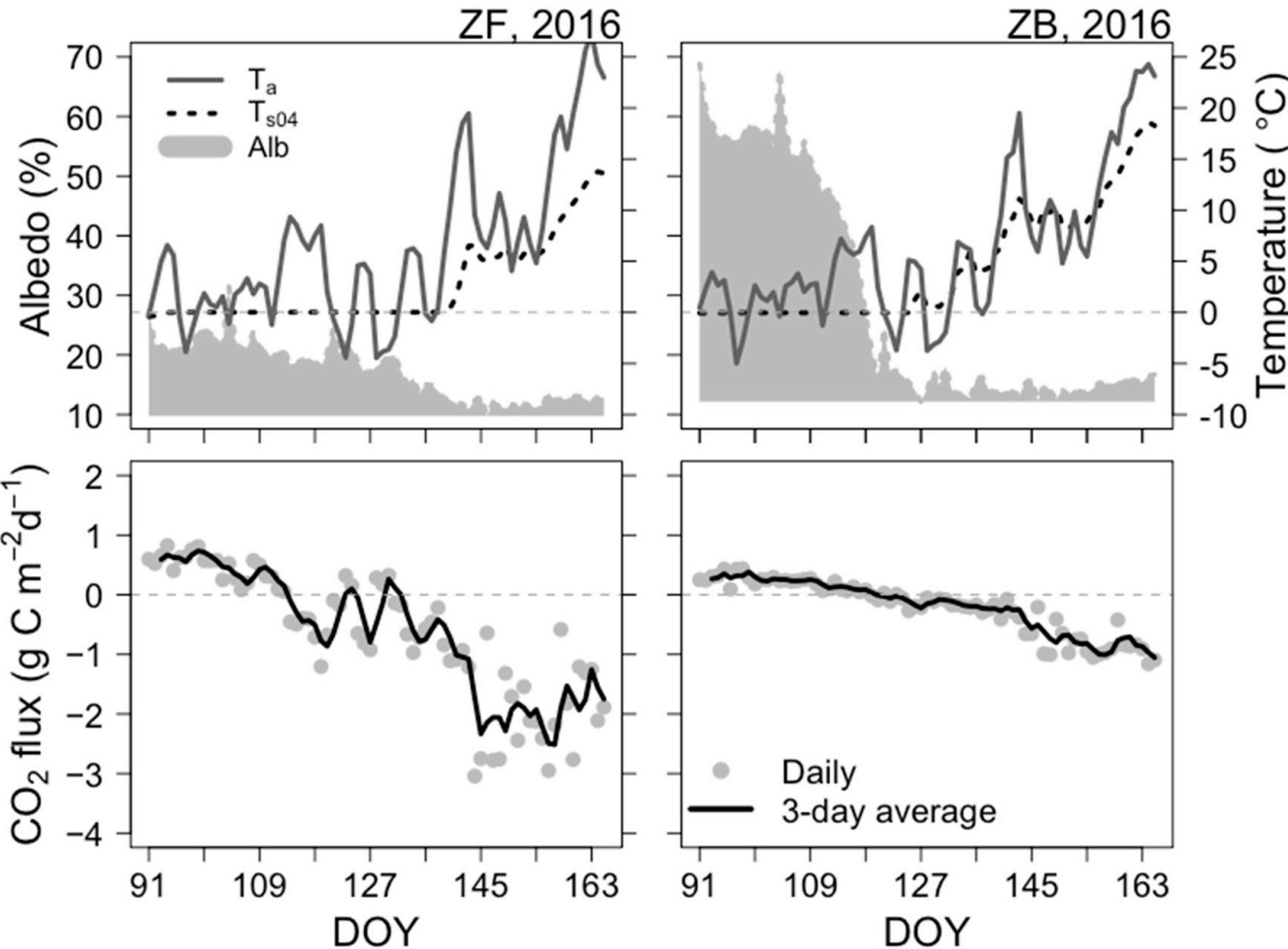

**Figure 2.** Time-series of three major environmental drivers and $CO_2$ flux for the period DOY 91–165 in 2016 at ZF and ZB sites. Upper panel: daily mean air temperature ($T_a$, dark-grey line; unit: °C), soil/peat temperature at 0.04 m ($T_{s04}$, dotted black line; unit: °C), and albedo (Alb, grey-shaded area; unitless). Lower panel: daily net $CO_2$ flux (grey circle; unit: g C $m^{-2}$ $d^{-1}$) and 3-day moving average $CO_2$ flux (black line; unit: g C $m^{-2}$ $d^{-1}$). The dashed grey line denotes the zero-line of $CO_2$ flux, with positive $CO_2$ flux values indicating a net $CO_2$ source and negative values indicating a net $CO_2$ sink.

In a typical springtime, $T_a$ was highly variable, ranging from −5 °C in April to 25 °C in May (Figure 2). The amplitude of $T_a$ was ~30 °C, which is typical of boreal spring weather. At the end of the snowmelt stage, the increase in $T_a$ was considerable (by ~15 °C). This may have led to the disappearance of snow cover on the ground surface, resulting in the observed increase in $T_{s04}$ during this period to above 5 °C. As shown in Figure 2, we also confirmed that $T_{s04}$ at ZB increased above 15 °C, which was ~5 °C warmer than at ZF for every spring in the study period (Supplementary Figure S1).

At both ecosystems, the start of net $CO_2$ uptake occurred during periods with cold or frozen soil (Figure 2). However, the magnitude of daily $CO_2$ uptake was lower than −1 g C $m^{-2}$ $d^{-1}$ until thawing of the surface soil ($T_{s04}$ < 0 °C) or snowmelt was complete. $CO_2$ uptake rates remained almost neutral and constituted a net $CO_2$ sink only when $T_{s04}$ recovered above 0 °C (DOY 120). As shown in Figure 2, data for both sites reveal that the transition from net $CO_2$ source to net $CO_2$ sink can be disrupted temporarily by cold spells (DOY 110–135). This feature was more distinct in forest ecosystem.

In 2016, the transition from net $CO_2$ source to net $CO_2$ sink occurred earlier in the forest than in the bog (Figure 2). Although the soil was still frozen, if the temperature was favourable (~5 °C) for photosynthesis, the forest ecosystem transformed into a weak net $CO_2$ sink. For instance, the start of net $CO_2$ uptake at ZF was on DOY 114 in 2016. Mean $T_a$ from DOY 113 to 119 was 7.7 °C, which favoured the triggering of the start of net $CO_2$ uptake. Compared with ZF, ZB was a not yet an active $CO_2$ sink, probably because the ground layer peat mosses were partially covered by snow.

We used the MARS model to examine the relative importance of factors controlling $CO_2$ fluxes over the study period. The fitting of this model showed fairly good predictive performance (84–89%; Table 3). For ZF, the MARS model identified six variables ($T_{s04}$, PAR, Alb, $T_a$, $T_{s16}$, and $T_{s64}$) as the important influences on $CO_2$ fluxes out of the nine analysed. The percentage of variance in daily $CO_2$ fluxes explained by the model ($R^2$) was 84%. For ZB, the MARS model identified five variables ($T_{s04}$, Alb, $T_{s64}$, PAR, and $T_{s32}$) as the important influences on $CO_2$ fluxes out of the seven variables analysed, with $R^2$ of 89% at ZB. Although we did not use ecological model, results show that the MARS model captured the controlling factors of $CO_2$ fluxes fairly well by considering non-linear relationships between modelled $CO_2$ fluxes and abiotic drivers.

**Table 3.** The order of important variables with statistics (nsubsets, GCV, and RSS; see details in Section 2.4) of $CO_2$ flux estimated by using the MARS model in spring period (DOY 99–DOY 155) from 2013 to 2017 at the ZF and ZB sites. For ZF, a total of 285 daily means of variables ($T_a$, $T_{s04}$, $T_{s08}$, $T_{s16}$, $T_{s32}$, $T_{s64}$, Alb, PAR, and VPD) were used as initial data. The percentage of variance in daily $CO_2$ fluxes explained by the model ($R^2$) was 84%. For ZB, a total of 216 daily means of variables were used for the period DOY 111–155 in 2013 and DOY 99–115 in 2014–2016. Peat temperatures at depths of 0.08 and 0.15 m were excluded for the initial data of the MARS model due to the long-term data gaps (>3 months) for 2014–2015. The percentage of variance in daily $CO_2$ fluxes explained by the model ($R^2$) was 89%.

| Site Variable | nsubsets | ZF GCV | RSS | Site Variable | nsubsets | ZB GCV | RSS |
|---|---|---|---|---|---|---|---|
| $T_{s4}$ | 11 | 100.0 | 100.0 | $T_{s4}$ | 12 | 100.0 | 100.0 |
| PAR | 19 | 59.0 | 60.5 | Alb | 11 | 46.1 | 48.7 |
| Alb | 9 | 44.8 | 46.9 | $T_{s64}$ | 9 | 29.5 | 33.0 |
| $T_a$ | 7 | 30.9 | 33.5 | PAR | 9 | 29.5 | 33.0 |
| $T_{s16}$ | 6 | 25.8 | 28.4 | $T_{s32}$ | 8 | 26.3 | 29.7 |
| $T_{s64}$ | 5 | 17.2 | 21.0 | | | | |

*3.2. Impacts of Cold Weather on Photosynthesis-Related Parameters*

During the springtime, $T_a$ was highly variable, with several apparent cold and warm phases (Figure 2). To examine the overall impacts of cold weather on $CO_2$ fluxes, we compared the photosynthesis-related parameters between the year having the most frequent cold spring weather days (2014) and the year having the least frequent cold spring weather days (2015) using the PAR-NEP relationship (Figure 3 and Table 4). Here, we defined cold spring weather as the day with a minimum daily $T_a < 0$ °C. For both sites, 2014 had the most frequent cold weather days (13 days for ZF and 15 days for ZB), and 2015 had the least frequent cold weather days (0 days for ZF and 2 days for ZB) compared with the entire study period (6 days for ZF and 10 days for ZB, 5-year mean values).

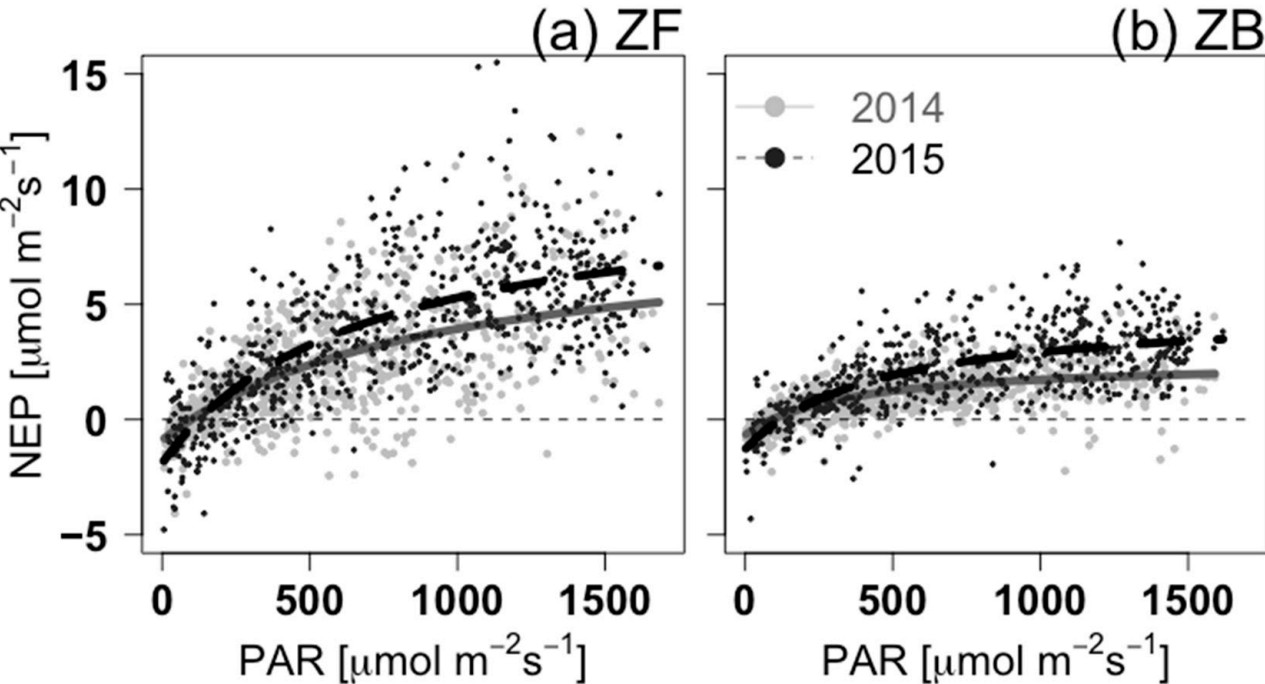

**Figure 3.** Ecosystem light-response curves for the year with the highest frequency of cold weather days (2014, grey dots and dark-grey lines) and the year with the lowest frequency of cold weather days (2015, black dots and dashed black lines) for (**a**) ZF and (**b**) ZB. Half-hourly daytime (potential global radiation > 20 W m$^{-2}$) data were chosen for after snowmelt (DOY 129–151 in 2014 and DOY 134–151 in 2015 for ZF; DOY 125–151 in 2014 and DOY 132–151 in 2015 for ZB). Fit parameters and statistics are listed in Table 4.

**Table 4.** Model parameters of the rectangular light-response function for the year with the most frequent cold weather days (2014) and for the year with the least frequent cold weather days (2015) for both ZF and ZB. Numbers in parentheses denote the standard errors of the parameters. $n$ is the number of half-hourly data. As NEP$_{sat}$ is a function of A$_{max}$, $\alpha$, R$_d$, and PAR at saturation PAR of 1500 $\mu$mol photon m$^{-2}$ s$^{-1}$, the standard deviations of NEP$_{sat}$ were taken as the averaged maximum and minimum standard errors of each parameter.

| Site | Year | $A_{max}$ [$\mu$mol m$^{-2}$ s$^{-1}$] | $NEP_{sat}$ [$\mu$mol m$^{-2}$ s$^{-1}$] | $\alpha$ | $R_d$ [$\mu$mol m$^{-2}$ s$^{-1}$] | $n$ | Residual Standard Error |
|------|------|------|------|------|------|------|------|
| ZF | 2014 | 9.22 (0.94) | 4.84 (0.70) | 0.010 (0.002) | −0.90 (0.21) | 761 | 1.83 |
| | 2015 | 11.95 (0.63) | 6.39 (0.76) | 0.018 (0.003) | −1.90 (0.36) | 659 | 2.12 |
| ZB | 2014 | 3.25 (0.14) | 1.96 (0.18) | 0.009 (0.002) | −0.68 (0.11) | 815 | 0.77 |
| | 2015 | 6.12 (0.28) | 3.41 (0.40) | 0.014 (0.002) | −1.30 (0.26) | 569 | 1.24 |

It is a typical feature that half-hourly $CO_2$ fluxes during spring daytime are highly scattered due to large temperature changes or cloud cover (Figure 3). Nevertheless, the light-response curves of both ecosystems showed typical shapes of rectangular hyperbola curve. The curve showed a rapid and linear increase in NEP at lower PAR levels, with a slow increase to reach maximum NEP at high PAR levels (Figure 3). Bogs fixed $CO_2$ at much lower PAR levels compared with forests. Under the same environmental conditions, for instance, needle-leaves at the top of the forest canopy are likely to warm sufficiently to photosynthesise compared with the understory vegetation, such as that found at the bog site. Due to the wider range of NEP for forest than those for bog, residual standard errors for ZF were larger than those for ZB (Table 4). Mean T$_a$ values after snowmelt in

2014 and 2015 were ~5 °C and ~13.5 °C, respectively. As a result, both ecosystems seem to have higher $A_{max}$ and $NEP_{sat}$ in 2015 than in 2014. $A_{max}$ from the rectangular hyperbola photosynthetic light-response function had a higher light saturation level than the one from the non-rectangular hyperbola curve because the latter had a stronger flattening at the light-compensation point. Therefore, $NEP_{sat}$ better represents values of the maximum $CO_2$ uptake rate of ecosystems. Compared with the bog, the forest had approximately three times higher $A_{max}$ and $NEP_{sat}$.

Quantum yields ($\alpha$) at both ecosystems were similar in 2014 and increased by more than 50% in 2015 (Table 4). In this period, the light-response curves for ZB reached at light-saturation point earlier than for ZF, indicating that the bog reached its maximum ability to fix $CO_2$ earlier than for the forest due to lower light-use efficiency (Figure 3). Interestingly, the relative changes in $NEP_{sat}$ of ZF and ZB in 2015 were approximately +32% and 74%, respectively. Presumably, the contribution of soil microbes in the ecosystem respiration term under the warm spring in 2015 may be greater than in 2014.

### 3.3. Interannual Variability of Spring Cumulative $NEE_{cum}$

Overall variations in spring $NEE_{cum}$ were more distinct at ZF compared with ZB, as shown by daily $CO_2$ fluxes in Figure 2. Spring $NEE_{cum}$ between 2014 to 2016 ranged from $-27.6$ to $-37.2$ g C m$^{-2}$ at ZF and from $-7.7$ to $-14.9$ g C m$^{-2}$ at ZB (Figure 4 and Table 5). Compared with the features at ZB, $NEE_{cum}$ at ZF showed larger amplitudes and higher uptake rates, which indicate that forest is a stronger spring $CO_2$ sink than the bog. When thick snow depth rapidly decreased with increasing temperature, between DOY 120 and 130, 2014–2016, both ecosystems released $CO_2$ in the atmosphere.

**Table 5.** Cumulative net ecosystem exchange of $CO_2$ ($NEE_{cum}$; g C m$^{-2}$), the day of year (DOY) of transition from $NEE_{cum}$ source to $NEE_{cum}$ sink ($NEE_{cum,tran}$; DOY), cumulative growing degree days of air temperature ($CGDD_{Ta}$; °C) corresponding to $NEE_{cum,tran}$, cumulative growing degree days of surface soil temperature ($CGDD_{T04}$; °C) corresponding to $NEE_{cum,tran}$, and mean slope ($\frac{\partial NEE_{cum}}{\partial DOY}$; g C m$^{-2}$ d$^{-2}$) after reading peak of $NEE_{cum}$ for ZF and ZB for the period DOY 99–155 in 2014–2016.

| Year | $NEE_{cum}$ (g C m$^{-2}$) | | $NEE_{cum,tran}$ (DOY) | | $CGDD_{Ta}$ (°C) | | $CGDD_{T04}$ (°C) | | $\frac{\partial NEE_{cum}}{\partial DOY}$ g C m$^{-2}$ d$^{-2}$ | |
|---|---|---|---|---|---|---|---|---|---|---|
| | ZF | ZB | ZF | ZB | ZF | ZB | ZF | ZB | ZF | ZB |
| 2014 | $-27.6$ | $-7.7$ | 128 | 138 | 104.5 | 164.6 | 0.3 | 41.7 | $-1.05$ | $-0.41$ |
| 2015 | $-37.2$ | $-14.9$ | 134 | 141 | 137.6 | 211.4 | 0.3 | 75.25 | $-1.62$ | $-0.92$ |
| 2016 | $-28.3$ | $-6.8$ | 126 | 142 | 83.0 | 141.9 | 0.85 | 55.42 | $-0.86$ | $-0.31$ |

To compare the rate of change of interannual variability in $NEE_{cum}$, the mean slopes of the curve ($\frac{\partial NEE_{cum}}{\partial DOY}$) following peak $NEE_{cum}$ were examined (Table 5). Both ecosystems showed steeper declining slopes after the peak of $NEE_{cum}$ in the three years studied, particularly the steepest in the warmest year (2015); $\frac{\partial NEE_{cum}}{\partial DOY}$ was $-1.62$ g C m$^{-2}$ d$^{-2}$ for ZF and $-0.92$ g C m$^{-2}$ d$^{-2}$ for ZB. For both ecosystems, 2015 was the year with the largest $NEE_{cum}$ for both the 3-year period (2014–2016) and the entire study period. In 2015, $T_a$ in May was ~10 °C, which was 4.2 °C higher than the 5-year mean (5.8 °C) (Figure 5). For both ecosystems, $NEE_{cum}$ was strong and negatively correlated with mean $T_a$ in May ($R^2 = 0.69$, $P = 0.081$ for ZF and $R^2 = 0.79$, $P = 0.110$ for ZB). Unusually high mean $T_a$ in May corresponded to the highest $NEE_{cum}$, meaning that the ecosystems were the strongest $CO_2$ sinks in the warmest year.

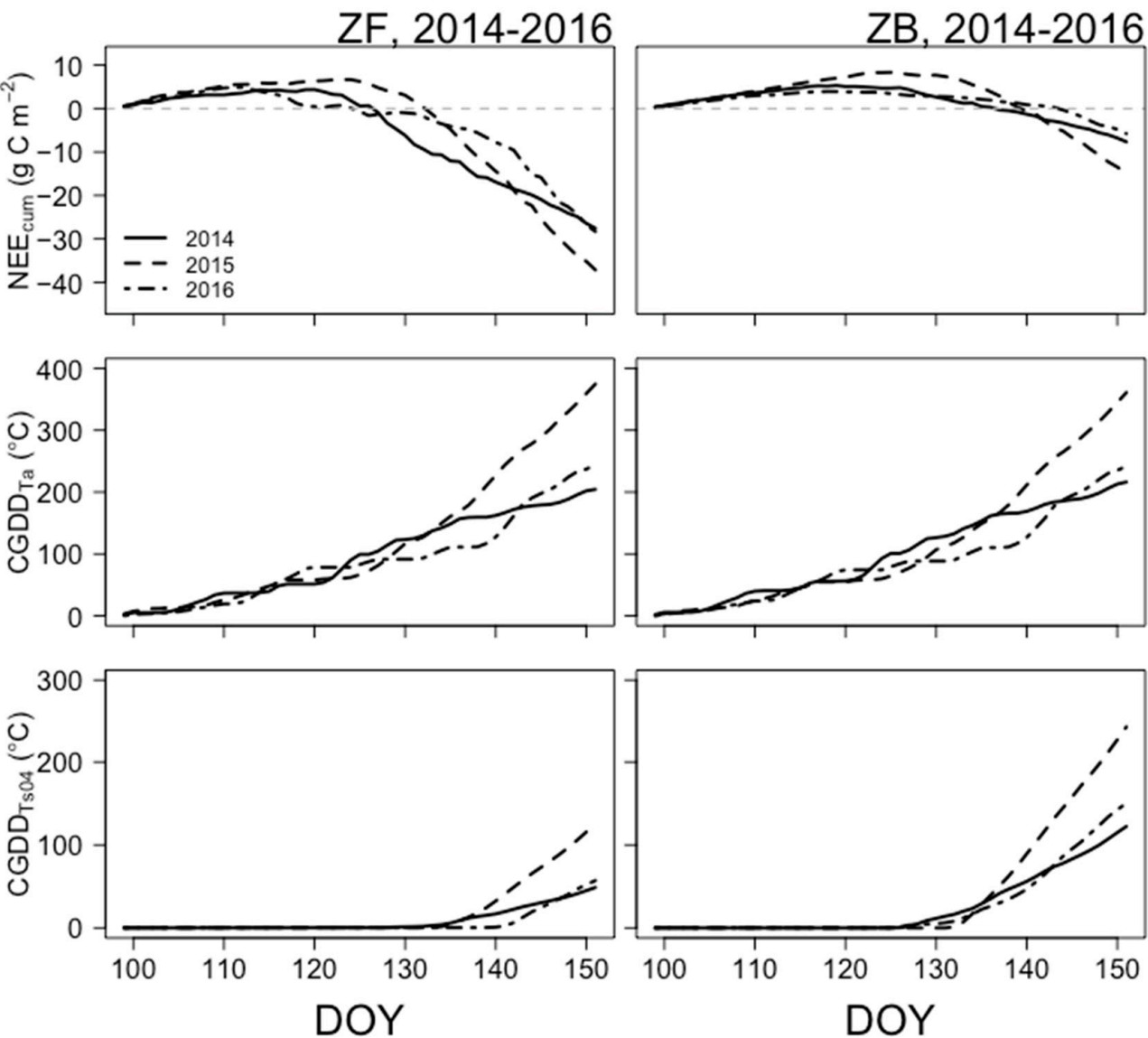

**Figure 4.** Cumulative net ecosystem exchange of $CO_2$ ($NEE_{cum}$; g C m$^{-2}$, **top**), cumulative growing degree days of daily mean air temperature ($CGDD_{Ta}$; °C, **middle**), and cumulative growing degree days of daily mean surface soil/peat temperature ($CGDD_{Ts04}$; °C, **bottom**) for ZF (**left** panel) and ZB (**right** panel). Solid, dashed, and long-dashed black lines denote data for DOY 99–155 for 2014, 2015, and 2016, respectively. Dashed grey zero-lines in the top panel denote the transition from a cumulative $CO_2$ source (positive, ecosystem carbon loss) to cumulative $CO_2$ sink (negative, ecosystem carbon gain).

Rapid increasing trends in accumulated growing degree days of air and soil temperatures were followed by the steep decreasing slope in $NEE_{cum,tran}$. These characteristics showed similar variations and trends over the three years. For instance, in 2014, transition day (DOY) from $NEE_{cum}$ source to $NEE_{cum}$ sink ($NEE_{cum,tran}$) at ZF was DOY 128, corresponding to $CGDD_{Ta}$ at 104.5 °C. At this point, $T_{s04}$ was positive but stayed close to 0 °C. This implies that $T_a$ plays an important role in $NEE_{cum}$ transition time, as we observed in Figure 2. Before surface $T_{s04}$ stays nearly 0 °C, the forest was net $CO_2$ source. Rapid increasing phase in $NEE_{cum}$ followed a steep and a linear increase of $CGDD_{T04} > 4$ °C. $CGDD_{T04}$ increased more than 7 °C in two days (DOY 135–137). After DOY 136, the slope of $NEE_{cum}$/DOY was $-1.05$ g C m$^{-2}$ d$^{-2}$. From this point, ZF entered the growing season, and photosynthesis had fully recovered from the cold season. Both photosynthesis and ecosystem respiration continuously increase over the growing season. At ZB, $NEE_{cum,tran}$

was 10 days later (DOY 138) than the $NEE_{cum,tran}$ at ZF. At this time, peats were already warmed up, showing positive temperatures ($CGDD_{Ta}$ = 164.6 °C). Similar to forest, daily mean $T_{s04}$ also increased approximately 7 °C in two days (DOY 128–130) immediately after $CGDD_{T04}$ exceeded 1 °C. Presumably, the ground surface at ZF is still partly covered by snow, and shading by trees may result in warmer surface conditions later than at ZB. If mosses were exposed to the open space without shading, then they could receive more direct solar radiation on the surface, which may result in earlier soil warm-up conditions. Characteristics of $NEE_{cum,tran}$ and heat accumulation explained by $CGDD_{Ta}$ and $CGDD_{T04}$ imply that the bog seems to have a higher level of temperature accumulation than forest, because soil microbial activity requires a suitable temperature level, corresponding to daily $T_{s04}$ > ~7 °C and $CGDD_{T04}$ of ~41 °C.

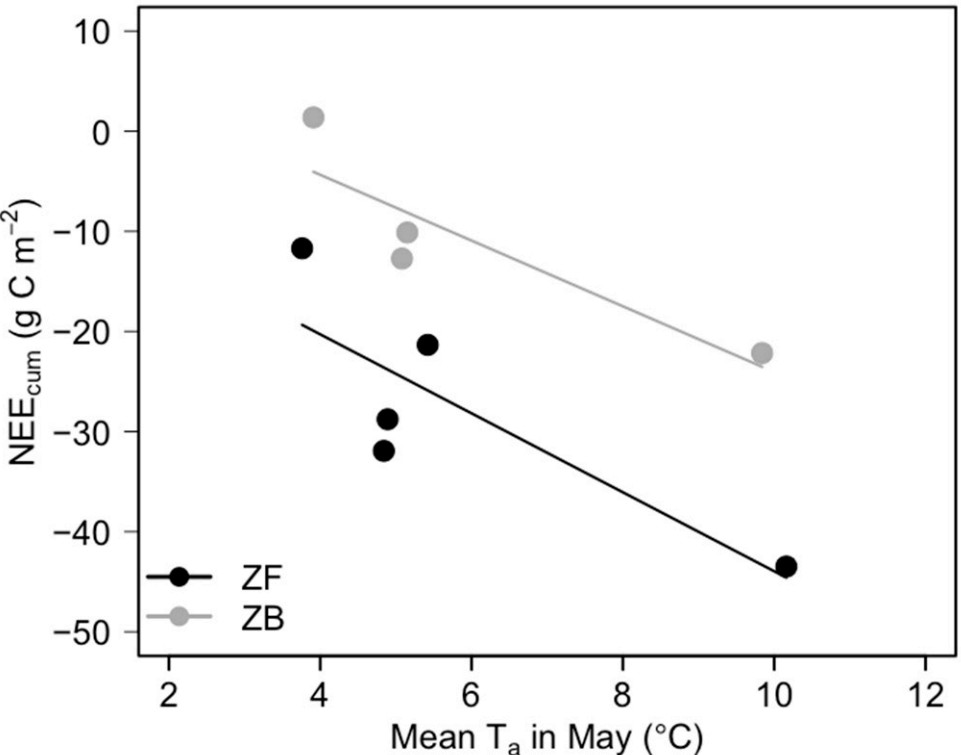

**Figure 5.** Relationships between cumulative NEE ($NEE_{cum}$; g C m$^{-2}$) and mean $T_a$ in May (°C) from 2013 to 2017 for the ZF and ZB sites. Linear regression for ZF (black dot and line): $NEE_{cum}$ = −4.52*$T_a$ − 3.94, $R^2$ = 0.69, $P$ = 0.081; linear regression for ZB (grey dot and line): 8.77*$T_a$ − 3.28, $R^2$ = 0.79, $P$ = 0.110. $NEE_{cum}$ was calculated from DOY 110–155 in 2013 only for ZB, and the rest of other years for both sites were calculated from DOY 99–155.

In 2015, $NEE_{cum,tran}$ at ZF occurred in DOY 134, corresponding to $CGDD_{Ta}$ at 137.6 °C. Although the spring mean $T_a$ in 2015 was the highest, $NEE_{cum,tran}$ in 2015 occurred the latest compared with other years. Perhaps, warm temperature conditions increase ecosystem respiration, leading to the forest remaining a net $CO_2$ source for longer. To find a reasonable answer, we will intend to examine components of the flux partitioning for a future study. Anomalously high $T_a$ also led to the highest $T_{s04}$. $CGDD_{Ta}$ during DOY 100–150 matched with the transition point of $CGDD_{T04}$ > ~4 °C. Compared with ZF, $CGDD_{T04}$ corresponding to $NEE_{cum,tran}$ at ZB varied 40–75 °C. $CGDD_{T04}$ at ZB showed more rapid enhancement in other years, possibly associated with the soil microbial responses to warm temperature conditions.

## 4. Discussion

### 4.1. Identifying the Major Drivers of $CO_2$ Fluxes

At our sites, spring $CO_2$ fluxes were mainly controlled by air temperature (Table 2). A rapid rise in $CO_2$ uptake occurred after surface snow melted, and $T_{s04}$ warmed above 1 °C (Figure 2). We confirmed that abiotic controlling factors of $CO_2$ fluxes identified from the MARS model agreed with previous findings. In boreal coniferous forests, spring photosynthesis recovery is related to air temperature, surface soil temperatures, first day of snowmelt, and final day of snowmelt [20,22–26,29]. The coniferous forest was already a net $CO_2$ sink before the end of snowmelt and while part of the soil surface was still frozen (Figure 2). In a wider context, this result agrees with the finding of Parazoo et al. [55], who showed that spring thaw was the crucial trigger for the start of spring photosynthesis and net $CO_2$ uptake in boreal forests in Arctic ecosystem. In northern peatland, peat temperature and snowmelt completion have been shown to affect the rapid increase in vegetation productivity [11,18,21,25]. In addition, PAR and temperature have been identified as controlling factors on the $CO_2$ flux of peat during the growing season [26,56,57].

In our study, $T_{s04}$ was identified as the primary driver of $CO_2$ fluxes for both ecosystems (Table 2). From a biophysical point of view, light (i.e., PAR) is known to be the primary driver of the beginning of net $CO_2$ sink or photosynthesis. In a previous study, a data-driven approach similar to the one used here identified PAR as the most important driver of $CO_2$ fluxes at the ZF site [42]. During the growing season, including the snow-melting period, PAR strongly controls the growth of *Sphagnum* [57]. In the boreal winter-to-spring transition period, snow cover decreases as temperature rises, as shown in reverse patterns of surface reflectance and air temperature [21–23]. This reflects that, in reality, $T_{s04}$ and Alb are indirect drivers of $CO_2$ fluxes.

We chose the MARS model to identify the relative importance of controlling factors of $CO_2$ fluxes because of its simplicity. In our study site, we often met a long-term data gap, particularly in the winter and early season, meaning that estimates of a reliable annual carbon budget with uncertainty are challenging. To overcome this limitation, further efforts for utilizing major variables identified from the MARS model and/or incorporating with other machine-learning methods suitable for a relatively smaller dataset (e.g., support vector machine) are desirable. With such efforts, we can develop an advanced version for the site-specific gap-filling algorithm.

### 4.2. Potential Drivers of Spring $CO_2$ Fluxes

We observed that $T_a$, $T_{s04}$, Alb, and PAR affect seasonal variations of $CO_2$ fluxes at a boreal forest and a bog. However, there is still room for exploring unknown or missing drivers of spring $CO_2$ fluxes. For instance, a recent study by Koebsch et al. [58] suggested that biological drivers seem more important for determining the variability in maximum gross primary productivity compared with abiotic drivers in boreal peatlands. Peichl et al. [59] also emphasized the important role of phenology on seasonal variation of photosynthesis in European peatlands. In addition, it is likely that the rapid increase in net $CO_2$ uptake after the snowmelt completion at ZB (Figure 2) may be associated with increases in peat temperatures in the rooting zone (0.1–0.2 m), leaf nitrogen, and chlorophyll *a* concentration [25]. This suggests that chlorophyll *a* could be a potential biochemical driver of spring $CO_2$ fluxes in peatlands.

In peatland, hydrological drivers (e.g., precipitation, soil moisture, and water-table depth) and their regime are also crucial for understanding photosynthesis-related processes. Previous studies [11,60,61] suggest that the water-table depth is an important control on the growing season NEE and the annual $CO_2$ balance, although this control appears to be site dependent. In contrast to previous studies, Strachan et al. [62] did not find significant relationships between peatland productivity or cumulative $CO_2$ exchange and early season temperature, the timing of snowmelt, or growing season length at an ombrotrophic bog located in Canada. Our data was limited in analyzing the roles of hydrological factors

on seasonality of $CO_2$ uptake in peatland; thus further efforts to examine the roles of biotic and abiotic drivers in regulating peatland C cycle at ZB are desirable. In addition, measurement of other important components of C flux, such as methane, are necessary to better understand the annual C balance in peatland [1,17,63,64].

*4.3. Role of Air and Soil Temperatures on Vegetation $CO_2$ Uptake Capacity in Northern Eurasia*

Consistent with characteristics of abiotic drivers of $CO_2$ fluxes shown in Figure 2, $NEE_{cum,tran}$ at ZF requires suitable air temperature accumulation ($CGDD_{Ta} > \sim 100\ °C$) more than soil temperature (Figure 4, Table 5). In previous studies, 5-day running mean of daily mean air temperature and cumulative temperature were good predictors of the commencement of spring photosynthesis in boreal coniferous forests [22,23].

We confirmed that both ecosystems have a zero-curtain period at $T_{s04}$ that remained close to 0 °C around until mid-April (Figure 2). This feature is a typical phenomenon in high-northern latitude or alpine ecosystems during the winter-to-spring transition period [21,23,25,60,65–67]. Although the surface snow may not have fully disappeared, warm spells may affect the timing of the start of net $CO_2$ uptake in boreal forests (Figure 2). Such phenomena may be a piece of evidence that Scots pines growing in high latitudes have adapted to water-limited environments in severe and long winters. Generally, low soil-temperature conditions during spring constrain water uptake and root activity in boreal conifers [24,66,68,69]. However, there is observational evidence that pine trees in winter use stem-stored water regardless of snowmelt termination or available soil water to maintain their metabolism [22,70]. It is also possible that different tree species may have different strategies for the process of spring photosynthesis recovery [71].

Cold weather appears to temporarily reduce $CO_2$ uptake rates over the study period. However, plants recovered $CO_2$ uptake rates as soon as $T_a$ increased above 0 °C (Figure 2). We found a similar pattern in other years (Supplementary Figure S1), which tend to be more distinct in the forest than in the bog. In addition, our data reveal that colder spring weather conditions resulted in a reduction of the maximum photosynthetic capacity (Figure 3). In previous studies, intermittent cold temperatures, for instance, frost events, delayed the photosynthesis recovery process of conifers, and the effect is more pronounced if frost is severe [20,72,73]. In peatlands, chilling can reduce chlorophyll *a* concentration [25,74]. It is unclear to find these features from daily $CO_2$ fluxes in our study; however, Figure 3 and Table 4 seem to show clues; cold weather temporarily suppresses photosynthetic capacity during spring. Overall, our data showed reasonable ranges and PAR-NEP fit parameters compared with the previous study [64].

A warmer climate may result in an earlier start of spring photosynthesis and increased vegetative productivity as well as annual $CO_2$ uptake in boreal coniferous forests [29,30,73]. However, the effect of warm temperatures on the net $CO_2$ uptake capacity of peatlands is likely to be more complex than that of forests because of soil hydrological conditions. Precipitation, water table depth, vegetation type, and phenology are recognised as other important controls on $CO_2$ fluxes, and their effects on net $CO_2$ uptake seem to be highly site dependent [11,12,14,37,59].

In this study, both ecosystems showed the highest vegetation productivity in 2015 (Figure 5). Despite the small amount of data and highly deviated data in 2015, both ecosystems showed negative relationships between $NEE_{cum}$ and mean $T_a$ in May. Based on the spatial distribution of 2m temperature anomalies over the study period (2013–2017), it is highly likely that vegetations in Zotino absorb the highest $CO_2$ uptake among the years because of anomalously warm temperature in May (Supplementary Figure S2). Anomalous warm temperatures were spread over the western Siberia, including the Zotino site. A study by Alekseychik et al. [37] seems to support that our finding is possible. They reported that the spring in 2015 was warmer; monthly mean air temperature in May was 4.1 °C higher than the long-term average in a similar type of bog in Finland. Besides, Liu et al. [14] and Pulliainen et al. [29] found that net $CO_2$ uptake in boreal ecosystem during the warm spring was greater than the normal year, implying that vegetation is

sensitive to the changes in air temperature, and it is a crucial driver to understand spring C uptake. To obtain the robust relationship between $NEE_{cum}$ and mean $T_a$ in May, we plan to analyze the longer term of vegetation indices using remote sensing data together with flux data. In addition, Alekseychik et al. [37] also found that despite the very warm spring in 2015, cloudy and rainy weather conditions in summer in that year resulted in a lower $CO_2$ uptake for the growing season compared with other years. This implies that the high vegetation productivity induced by the anomalously warm temperatures in May 2015 is likely to have been compensated by the cool and wet ensuing summer. Our forthcoming research will examine the annual net $CO_2$ balance through comparison of the results of the present study with those from other sites in Siberia and for different weather conditions.

## 5. Conclusions

$CO_2$ flux observations at the Zotino sites showed that distinct differences exist in flux magnitude, the timing of start of net $CO_2$ uptake, and the rate of $CO_2$ uptake between the boreal forest and bog. In spring, the boreal forest generally changed from net $CO_2$ source to net $CO_2$ sink approximately 1–2 weeks earlier than the bog. We found similar aspects for the transition from $NEE_{cum}$ source to $NEE_{cum}$ sink. Rapid net $CO_2$ uptakes for both ecosystems covaried with $T_a$, corresponding to $T_{s04}$ values $> 5$ °C. To change into complete NEEcum sink, ZF seems to require CGDDTa of ~80 to 137 °C and ZB requires CGDDTa of 141 to 211 °C. Compared with ZF, seasonal variations in NEE-cum at ZB were smaller. This implies that forests appear more sensitive to the changes in air temperature than bogs under the same spring weather conditions. We confirmed that cold spring weather reduced the maximum photosynthetic capacity in both ecosystems. During the study period, abnormally warm air temperatures in spring 2015 resulted in the highest vegetation productivity. Overall, results suggest that the $CO_2$ uptake capacities of boreal forest and bog are sensitive to rising air temperature in springtime. This strong relationship between air temperature and NEE seems to support that a linear relationship between NEE estimated from the atmospheric inversion method and air temperature proposed by Rödenbeck et al. [75].

Our analysis is limited in analyzing NEE; thus, future work will examine the effects of anomalously warm spring weather on both photosynthesis and respiration with respect to the annual C balance. Besides, we will investigate the linkage between hydrological conditions (e.g., snow depth and rainfall) in the previous winter and seasonal/annual $CO_2$ balances in the following year. To overcome the limited number of EC flux measurement, remote-sensing-based phenology and photosynthesis products will be utilized. We anticipate direct $CO_2$ flux measurements at Zotino will be valuable for evaluating biogeochemical models. Further, it will provide insights into the prediction for the future terrestrial C cycle in northern Eurasia, particularly in remote and relatively undisturbed natural ecosystems.

**Supplementary Materials:** The following are available online at https://www.mdpi.com/article/10.3390/atmos12080984/s1, Figure S1: Same as Figure 2, but for (**a**) 2013, (**b**) 2014, (**c**) 2015, (**d**) 2017 at ZF and ZB. No available $CO_2$ flux measurements at ZB in 2017, Figure S2: Spatial distribution of 2 m temperature anomalies in May for each year over the 5-year mean (May, 2013–2017) from Climate Reanalyzer (https://ClimateReanalyzer.org, accessed on 17 July, 2021), Climate Change Institute, University of Maine, USA. ERA5 is the fifth generation ECMWF atmospheric reanalysis data produced by the Copernicus Climate Change Service (C3S). The spatial resolution of ERA5 is at 0.5°. ZOTTO is situated approximately at 60°N, 89°E. 2m temperature anomaly for (**a**) May 2013, (**b**) May 2014, (**c**) May 2015, (**d**) May 2016, and (**e**) May 2017, respectively.

**Author Contributions:** Conceptualization, S.-B.P., A.K., and M.M.; methodology, S.-B.P., M.M., T.T., I.M., O.P. and A.P.; software, S.-B.P. and O.K.; validation, S.-B.P. and O.K.; investigation, S.-B.P., A.K. and M.M.; resources, T.V.; writing—original draft preparation, S.-B.P.; visualization, S.-B.P.; supervision, A.K.; funding acquisition, J.L., M.H. and S.S.P. All authors have read and agreed to the published version of the manuscript.

**Funding:** The ZOTTO project is funded by the Max Planck Society through the International Science and Technology Center (ISTC) partner project no. 2757 within the framework of the proposal "Observing and Understanding Biogeochemical Responses to Rapid Climate Changes in Eurasia". S.-B.P. and S.S.P. are supported by National Research Foundation of Korea (NRF- 2020R1C1C1013628). A.P. is supported by grant RFBR #18-05-60203-Arktika. T.V. thanks the grant of the Tyumen region, Russia, Government in accordance with the Program of the World-Class West Siberian Interregional Scientific and Educational Center (National Project "Nauka").

**Institutional Review Board Statement:** Not applicable.

**Informed Consent Statement:** Not applicable.

**Data Availability Statement:** Data are available upon reasonable request to the corresponding author.

**Acknowledgments:** S.-B.P. acknowledges the International Max Planck Research School for Global Biogeochemical Cycles (IMPRS-gBGC). We deeply thank the technical staff for maintaining the eddy covariance flux towers: Karl Kübler, Steffen Schmidt, and Martin Hertel from the Freiland group, Max Planck Institute for Biogeochemistry in Jena; Alexey Panov, Alexander Zukanov, Nikita Sidenko, Sergey Titov, and Anastasiya Urban from the V.N. Sukachev Institute of Forest in Krasnoyarsk; and many other supporters in Zotino. S.-B.P. thanks to colleagues at the MPI-BGC; Andrew Durso, Kendalynn Morris, Jeffrey Beem-Miller, and Shane Stoner for proofreading on the manuscript; Talie Musavi for advising statistical analysis; and Marcus Guderle for creating Figure 1. Authors are grateful to three anonymous reviewers for their helpful comments and suggestions for improvements of the manuscript.

**Conflicts of Interest:** The authors declare no conflict of interest.

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
