# Peer review of "Temperature Control of Spring CO2 Fluxes at a Coniferous Forest and a Peat Bog in Central Siberia"

_atmosphere, doi:10.3390/atmos12080984_

Round 1
Reviewer 1 Report
This study reports on spring time (April to mid-June) net ecosystem exchange in a forest and bog in boreal Siberia over a 3-5 year period. The authors report that warmer springs lead to greater net CO2 uptake, but this response to warming is greater in the forest than bog.
Overall the study is well done and clearly written. I note some small corrections at the end in minor corrections. There are some more substantial corrections needed as well prior to publication. Firstly, the study site description for the bog site appears to be missing and must be added. Secondly, the regression between cumulative NEE in spring and the mean May air temperature is not fully convincing. The regression lines are not significant for either the bog or forest at p<0.05 and the forest appears largely driven by 1 data point. While there is still an overall pattern that is quite convincing, this limitations need to be pointed out clearly in the results and discussion.
Additional comments:
Lines 37-38: it was not clear to me where these accumulated temperature sums came from as I don't recall seeing them in the main results. If I'm wrong, nothing to change, but just make sure these are also clearly stated in the main body of the manuscript.
Line 39: ZF has not yet been define, so I suggest saying forest here.
Line 105: In objective a) it's not clear what difference you're referring to, difference between bog and forest? among years?
Lines 178-179: It would be useful to also give the dates here to help the reader interpret the DOY.
Line 266: I found the use of the acronym SCU very confusing in the text and don't really think it is necessary. I suggest just saying the start of net CO2 uptake when needed instead of SCU.
Line 278: I think ground-attached peat mosses sounds awkward. I suggest changing "relatively short and ground-attached" to "ground layer"
Figure 3: This caption is incorrect. It is the same as Figure 2 and does not describe this figure. Revise.
Line 340-341: I wasn't clear on what light response curves flattened earlier really meant. Do you mean that NEP leveled at a lower PAR? I think this needs to be clarified.
Line 345-346: The effect of soil temperature on soil respiration is pretty well-established, so I found this general statement odd. Can you make this more specific and highlight exactly what gap needs to be addressed? Maybe the effect of temperature in springtime?
Line 356-358: I agree that the change in NEEcum at ZF is bigger than ZB, but I was initially confused by the following sentence where the interannual variation given is much bigger at ZB than ZF (and I think that is just due to the years reported as the overall variation in Figure 5 is also bigger for ZF. I suggest revising this section to avoid confusing the reader.
Lines 388-389: I didn't follow the logic here. More microbes cause the soil to warm up faster? I would guess it has more to do with moisture content affecting soil thermal properties and maybe shading by the canopy in the forest. I suggest removing this sentence and updating with other hypotheses for the difference.
Table 5: I suggest giving the time period over which NEEcum is computed in the table caption. Also, do you really think 2 decimal places is an appropriate level of precision? I would guess one decimal or even rounded to the whole number for NEEcum is more appropriate.
Line 535: I'm not clear on what NEE-T inversion means. Please provide more information here.
Minor corrections:
LIne 72: flux not fluxes
LIne 78: Should stat be start here?
Line 80: remove (have shown??)
Line 99: remove one of the "measurements" from this sentence.
Line 200: Remove were after variables
Line 227: NEP as the same of -NEE doesn't really make sense. Revise.
Line 233: Change "of" to "to"
Line 428: rise not rising
Line 429: fix the degree symbol in 1 oC
Line 437: have not haven
Line 450: I think this should be long-term data gap (insert the word data)
Lines 491-492: severe and long winters is repeated in this sentence
LIne 520: ensuing is repeated twice in this sentence
Line 531: weather not weathers
Line 540: change limited in ground-based to limited number of ground-based
LIne 542: I suggest changing "is" to "will be"
Author Response
We sincerely appreciate the time and effort that the reviewers have dedicated to providing valuable feedback on our manuscript. We have been able to incorporate changes to reflect most of the suggestions provided by the reviewers. We have highlighted the changes within the manuscript. Further corrections by authors have been made through the revision process and new line numbers were shown for each response. Please see the attachment.

Reviewer 2 Report
Please refer the attached PDF.

Author Response

(The authors gave the same response as above.)

Reviewer 3 Report
Atmosphere-1288027
Temperature control of spring CO2 flux at a coniferous forest and a peat bog in central Siberia
Park et al. present an analysis of several years of CO2 flux data from eddy-covariance towers at a remote forest and bog site in Siberia, with additional data regarding soil temperatures and the timing of snowmelt and complete disappearance of snow in each spring. They concentrate on the effects of variability in temperature and snowmelt in the winter-spring transition period, when these ecosystems switch from a net source of CO2 to atmosphere to a net sink as photosynthesis increases rapidly.
Overall
Well written, mostly quite clear. A few minor ambiguities could be easily corrected with careful proofreading, for example the paragraph from LN 470 to LN 478 is somewhat unclear regarding the patterns found in this study, compared to what was reported in [66]; the use of citations as nouns in sentences contributes to this ambiguity.
This study represents an important step in filling a serious knowledge gap regarding drivers of annual variation in CO2-exchange in remote boreal regions, and certainly would help to provide a solid foundation for further investigations.
Introduction
LN 76 – “A study by [29] showed” please change to “A study by Pulliainen et al. [29] showed”
LN 80 – I prefer “have shown” over “showed” in this sentence
LN 81 – please change to “A slowed temperature change may”
LN 103 – please change to “study of Arneth et al. [20] in”
Please implement this change throughout the paper – where you write [number] as the subject or object of the sentence (e.g. study by [29]) please change it to the author or authors names with the number in square brackets immediately following.
Materials and Methods
Figure 1 – how big are the pixels in the upper part of this figure? Is a single square pixel a few metres wide?
LN 118-119 – are there no trees younger than 80 years at the site?
LN 194 – the program R and the package “earth” are not cited correctly. They can be cited in the usual way (numbers in square brackets in text, details in References section); for the full information, in R enter
> citation()
> citation(earth)
Discussion
LN491-492: redundant use of “severe and long winters”. Delete the second use of these words.
Typos and other trivial errors
LN 78 – “stat” becomes “start”
LN 149 – the -1 after min should be superscript
Author Response

(The authors gave the same response as above.)
